# Predicting the Impact of Climate Change on the Future Distribution of *Paederus fuscipes* Curtis, 1826, in China Based on the MaxEnt Model

**DOI:** 10.3390/insects15060437

**Published:** 2024-06-09

**Authors:** Hui Gao, Xinju Wei, Yaqin Peng, Zhihang Zhuo

**Affiliations:** 1College of Life Science, China West Normal University, Nanchong 637002, China; hubeihuige@cwnu.edu.cn (H.G.); weixinjuxx@foxmail.com (X.W.); pengyaqin2023@foxmail.com (Y.P.); 2College of Environmental Science and Engineering, China West Normal University, Nanchong 637002, China

**Keywords:** *Paederus fuscipes*, MaxEnt, potentially suitable distribution, environmental variables, climate change, centroid

## Abstract

**Simple Summary:**

*Paederus fuscipes* belongs to the Coleoptera order, Staphylinidae family, and *Paederus* genus, and has a wide distribution. It is a predatory insect and serves as a predatory natural enemy of agricultural and forestry pests. Additionally, it acts as a vector for Paederus dermatitis in most tropical and subtropical countries worldwide. In this study, we utilized the MaxEnt model and ArcGIS software to predict the current and future suitable habitat distribution of *P. fuscipes*, considering known distribution information, and climate and environmental factors. The results indicate that high suitability areas for *P. fuscipes* in China are mainly concentrated in the Shandong Hills, the North China Plain, and the middle and lower reaches of the Yangtze River Plain. Predictions based on future climate change scenarios show a significant decrease in high and low suitability areas while the moderately suitable area increases (except for the SSP2-4.5 scenario in the 2090s). This study promotes the use of *P. fuscipes* as a biological control agent against other pests and provides essential references for its application in agricultural and forestry pest control.

**Abstract:**

*Paederus fuscipes* Curtis, 1826, belongs to the Coleoptera order, Staphylinidae family, and *Paederus* genus (Fabricius, 1775). It has a wide distribution and strong invasive and environmental adaptation capabilities. As a predatory natural enemy of agricultural and forestry pests, understanding its suitable habitat is crucial for the control of other pests. This study, for the first time, uses the MaxEnt model and ArcGIS software, combining known distribution information of *P. fuscipes* and climate environmental factors to predict the current and future suitable habitat distribution of this insect. The key environmental variables affecting the distribution of *P. fuscipes* have been identified as mean diurnal range (mean of monthly (max temp-min temp)) (bio2), isothermality (Bio2/Bio7) (*100) (bio3), minimum temperature of the coldest month (bio6), temperature annual range (bio5-bio6) (bio7), mean temperature of the driest quarter (bio9), mean temperature of the coldest quarter (bio11), precipitation of the wettest month (bio13), precipitation of the driest month (bio14), and precipitation seasonality (coefficient of variation) (bio15). The highly suitable areas for *P. fuscipes* in China are mainly distributed in the hilly regions of Shandong, the North China Plain, and the middle and lower reaches of the Yangtze River Plain, with a total suitable area of 118.96 × 10^4^ km^2^, accounting for 12.35% of China’s total area. According to future climate change scenarios, it is predicted that the area of highly and lowly suitable regions will significantly decrease, while moderately suitable regions will increase (except for the 2090s, SSP2-4.5 scenario). These research findings provide important theoretical support for pest control and ecological conservation applications.

## 1. Introduction

*Paederus fuscipes* Curtis, 1826, belongs to the Coleoptera order, Staphylinidae family, and *Paederus* genus (Fabricius, 1775). It is a predatory insect and a natural enemy of agricultural and forestry pests. It also serves as a vector for *Paederus* dermatitis, which is prevalent in most tropical and subtropical countries worldwide. *Paederus* dermatitis, also known as linear dermatitis, is a skin condition caused by the rupture of nymph bodies. When their body fluids come into contact with human skin, they can cause acute erythematous vesicular lesions [1,2]. This insect has a wide distribution and is found across the globe outside of Antarctica, demonstrating strong invasive and environmental adaptation capabilities [3,4]. It inhabits moist environments such as riverbanks, swamps, and irrigated fields [5]. The generation number of *P. fuscipes* is related to temperature, humidity, and food availability. In regions like Jiangsu, Guangxi, and Anhui in China, three generations occur. They overwinter as adults, but in some areas, their overwintering is not during the winter; rather, their activity and population fluctuations seem to be caused by low humidity in the dry season [1,6]. They have a wide habitat range, often feeding on agricultural pests. Therefore, larvae and pupae of other insects become prey for rove beetles [7]. *P. fuscipes* can prey on various agricultural pests, such as aphids, leafhoppers, and planthoppers, and thrip on crops like rice, wheat, cotton, corn, and legumes, playing a significant role in biological pest control [8].

Species distribution models (SDMs) are empirical methods used to quantify the ecological niche of a species in its environment. By combining occurrence data samples of the target species with environmental variables from sample locations, SDMs infer the relationship between species and environment, thereby predicting the potential distribution of the species [9,10,11]. Commonly used niche models include CLIMEX, BIOCLIM, GARP, DOMAIN, and MaxEnt [12]. Among these models, the MaxEnt model is widely applied in various fields due to its advantages of using small sample sizes for prediction, high accuracy, fast computation speed, and simple operation [13,14]. Spatial data modeling has become an increasingly important tool and is widely used in ecology, conservation, evolutionary biology, geography, and climate change research [15,16]. Its applications include assessing the potential impact of climate on species [17,18], predicting the potential distribution of endangered plants [19], evaluating invasive species [20,21], and identifying and assessing flood indices and risks in urban areas [22], among others.

Currently, there is relatively little research on *P. fuscipes* in China. Existing studies mainly focus on its biological characteristics [6], chemical composition [7], predatory behavior [23], and the inhibitory mechanisms of plant essential oils on *P. fuscipes* [24]. There is limited research on the potential suitable habitats for *P. fuscipes*.

This study analyzed the environmental suitability of *P. fuscipes* based on known distribution data and Chinese environmental data, using the MaxEnt model and ArcGIS technology. We predicted the current and future potential distribution of *P. fuscipes* and analyzed the trend of centroid shift in its potential geographic distribution, providing a theoretical basis for pest control.

## 2. Materials and Methods

### 2.1. Species Distribution Data

The key to establishing an ecological niche model is the availability of sufficient known species distribution points [25]. The occurrence data for *P. fuscipes* mainly come from relevant literature, accessed through the China National Knowledge Infrastructure (CNKI, https://www.cnki.net/, accessed on 19 February 2024), combined with data from the Global Biodiversity Information Facility (GBIF, https://www.gbif.org/, accessed on 19 February 2024). The specific latitude and longitude information for species distribution points is determined using Google Earth (http://www.earthol.com/, accessed on 19 February 2024). The collected distribution points of *P. fuscipes* are imported into ArcGIS software 3.4.4, and a buffer analysis method is used to filter the obtained distribution points, excluding overfitting caused by strong spatial correlations. The resulting records are saved in “CSV” format. A total of 210 occurrence records were obtained, as shown in Figure 1.

### 2.2. Environmental Variables and Data Processing

Environmental variables are crucial factors that influence species distribution [25]. Therefore, in this study, we downloaded 19 bioclimatic variables from the Worldclim dataset (http://www.worldclim.org/, accessed on 19 February 2024). For future climate data, we selected three scenarios from the new generation climate system model BCC-CSM2-MR developed by the Beijing Climate Center (BCC): SSP1-2.6 (low greenhouse gas emission scenario), SSP2-4.5 (medium greenhouse gas emission scenario), and SSP5-8.5 (high greenhouse gas emission scenario) to predict the distribution of *P. fuscipes*. Due to the strong autocorrelation among environmental variables and the fact that not all environmental variables are necessary when predicting species distribution [26,27], we used the Pearson correlation coefficient (r) to identify multicollinearity (Table 1). Environmental variables with a correlation coefficient greater than or equal to 0.8 were removed to reduce multicollinearity and model overfitting, while variables with a correlation coefficient less than 0.8 were retained. We then reconstructed the distribution model using these key environmental variables.

### 2.3. The Modeling Process

To explore the impact of environmental factors on *P. fuscipes*, we combined distribution point data for *P. fuscipes* with selected environmental variables. Using MaxEnt 3.4.4 software, we simulated the potential distribution of *P. fuscipes*. To reduce uncertainty caused by the random selection of data, 75% of species occurrence data was randomly selected as the training dataset, while the remaining 25% of species data was used as the testing dataset. The training process was repeated 10 times, with other parameters set to their default values [28,29]. We used jackknife analysis to determine the contribution rates of key environmental variables and then derived the logistic relationship between distribution probability and climate factors through response curves [30].

We used ArcGIS software to extract the distribution areas of *P. fuscipes* in China and to analyze the climatic suitability for this insect. The output range simulated by MaxEnt software ranges from 0 to 1, where values closer to 1 indicate a higher likelihood of species presence [31]. Based on the Intergovernmental Panel on Climate Change (IPCC) likelihood report divisions for assessment methods [32], combined with the actual situation of *P. fuscipes*, its suitable habitat suitability is classified into four levels and represented in different colors: unsuitable areas (*p* < 0.1, white), low suitability areas (0.1 ≤ *p* < 0.3, yellow), moderate suitability areas (0.3 ≤ *p* < 0.5, orange), and high suitability areas (0.5 ≤ *p*, red). Using the SDM Toolbox 2.0 in ArcGIS, we calculated the centroid position of *P. fuscipes* in different periods to determine the migration direction of the distribution area of *P. fuscipes* while using the centroid distribution to evaluate the migration of its suitable habitat.

### 2.4. Model Evaluation

The Receiver Operating Characteristic (ROC) curve is used to assess the accuracy of model predictions, with the Area Under the Curve (AUC) under the ROC curve serving as a performance indicator for MaxEnt predictions [33]. The AUC ranges from 0.5 to 1, with values closer to 1 indicating higher model accuracy [34]. An AUC value of 0.5–0.6 indicates failed performance; 0.6–0.7 indicates poor performance; 0.8–0.9 indicates excellent results; and values above 0.9 indicate high performance.

## 3. Results

### 3.1. Model Performance and Key Environmental Variables

The accuracy assessment of the MaxEnt model’s predictions yielded an AUC value of 0.968, as shown in Figure 2. This indicates a high level of accuracy in the model’s predictions, providing a reliable basis for determining the potential distribution of *P. fuscipes* in China.

Combining the contribution rate and Pearson correlation coefficient, we selected nine key environmental variables and reconstructed the species distribution model. Table 2 shows the contribution percentages and ranking importance of the nine key environmental variables affecting *P. fuscipes*: mean diurnal range (mean of monthly (max temp-min temp)) (bio2), isothermality (Bio2/Bio7) (*100) (bio3), minimum temperature of the coldest month (bio6), temperature annual range (bio5-bio6) (bio7), mean temperature of the driest quarter (bio9), mean temperature of the coldest quarter (bio11), precipitation of the wettest month (bio13), precipitation of the driest month (bio14), and precipitation seasonality (coefficient of variation) (bio15). Bio14 had the highest contribution rate at 27.8%, followed by bio3 (18.1%) and bio13 (17%). The cumulative contribution rate of these key environmental variables reached 100%. This indicates that the above nine environmental variables represent crucial information for simulating the potential geographic distribution of *P. fuscipes*.

### 3.2. The Potential Distribution of P. fuscipes in the Current Period

According to the optimized MaxEnt model, a distribution prediction map of the most suitable habitats for *P. fuscipes* has been established (Figure 3). The results indicate that, under the current climate conditions, the highly suitable areas are mainly distributed south of the Sichuan Basin, north of the Yunnan-Guizhou Plateau, in the southeast hills, in the middle and lower reaches of the Yangtze River Plain, in Hainan Province, and the northeastern direction of Taiwan, mainly concentrated in the southern regions of China. In addition, a small portion of highly suitable areas is distributed southward of the Yarlung Zangbo River and east-southeast of the Tarim Basin in Xinjiang. Table 3 presents statistical data on the highly suitable areas for *P. fuscipes* in China under the current climate conditions. The results indicate that the highly suitable areas are mainly concentrated in provinces such as Guizhou, Hunan, Guangxi, Guangdong, Jiangxi, Fujian, Zhejiang, and Hubei. The province with the largest area of highly suitable areas currently is Hunan, with 19.27 × 10^4^ km^2^, followed by Guangxi with 17.22 × 10^4^ km^2^, and Guangdong with 15.38 × 10^4^ km^2^, accounting for 16.20%, 14.48%, and 12.93% of the total highly suitable area in the country, respectively. It is worth noting that the highly suitable area in Hunan accounts for 90.97% of the entire province, while in Shanghai, it accounts for 93.14% of the city. Additionally, the highly suitable areas in Fujian, Guangdong, and Hong Kong all exceed 80% of the total area of the province (or special administrative region). The distribution of moderately to highly suitable areas is closely related and widely distributed.

### 3.3. The Potential Distribution of P. fuscipes in the Future Period

Figure 4 depicts the predicted suitable habitats of *P. fuscipes* in China during the 2050s (2041–2060) and 2090s (2081–2100) under climate change scenarios SSP1-2.6, SSP2-4.5, and SSP5-8.5. Compared to the current distribution map, the high suitability areas for *P. fuscipes* significantly decrease in future climate scenarios, mainly concentrated in the Shandong Hills, North China Plain, and the middle and lower reaches of the Yangtze River Plain. The areas of moderate and low suitability show varying changes, but both are mainly concentrated in the southeast hills, the middle and lower reaches of the Yangtze River Plain, the Sichuan Basin, the Tarim Basin, and the North China Plain. Combining the data from Table 4, regardless of whether it is the 2050s or 2090s, the area of high suitability for *P. fuscipes* decreases sharply compared to its current distribution. In the 2090s, under the SSP1-2.6 scenario, the high suitability area decreased to 2.19 × 10^4^ km^2^, representing a 98.16% decrease compared to the current high suitability area for *P. fuscipes*. Similarly, under the SSP2-4.5 scenario in the 2090s, the area decreases by 94.84%. The area of moderately suitable habitats shows a slight decrease only under the SSP2-4.5 scenario in the 2090s, while in other scenarios, it increases to varying degrees, ranging from 24.38% to 71.22%. The most significant increase occurs in the 2050s under the SSP1-2.6 scenario, reaching 86.87 × 10^4^ km^2^, representing a 71.22% increase. In contrast to the significant changes in the high and moderate suitability areas, the low suitability area decreases in all scenarios, with a decrease ranging from 10.78% to 38.60%. Considering all scenarios for high, moderate, and low suitability areas, both the 2050s and 2090s show a significant reduction in suitable habitats, indicating a weakening trend in the future suitability of *P. fuscipes*.

### 3.4. The Environmental Variables Influencing the Geographical Distribution of P. fuscipes

Using jackknife analysis of environmental variables, we investigated the extent to which these variables impact the distribution of *P. fuscipes* (Figure 5). These key environmental variables all have some degree of influence on the potential distribution of *P. fuscipes*. The blue bands correspond to “with only variable,” indicating the importance of environmental variables in influencing species distribution. Bio11 (mean temperature of the coldest quarter) is the most important environmental variable affecting the distribution of *P. fuscipes*. Other variables with a regularized training gain exceeding 0.8 include bio6 (the minimum temperature of the coldest month), bio9 (the mean temperature of the driest quarter), and bio14 (the precipitation of the driest month). The green variables represent the uniqueness of the variables, with the length of the band negatively correlated with the specific information contained in the variable. Among these nine variables, precipitation of the wettest month (bio13) has the shortest green band, indicating that is important and contains unique information not found in other variables.

Based on the response curves of the *P. fuscipes* probability distribution depicted in Figure 6, the range of future distribution variables for *P. fuscipes* has been determined. The suitable temperature range for bio11 (Figure 6a) is −2.83 °C to −13.32 °C, with an optimal temperature of 6.06 °C. The curve shows a steep rise and fall, indicating the limited adaptability of *P. fuscipes* under low-temperature conditions. For bio6 (Figure 6b), the suitable temperature range is −7.30 °C to −9.33 °C, with the optimal temperature value being −1.84 °C. Unlike bio11, this variable shows a slow decline followed by a rapid decrease on the curve. It is worth noting that bio9 (Figure 6c) exhibits two suitable ranges within the selected temperature range. The first one starts rising from 0 at −11 °C, peaks at 3.37 °C, and then declines. Subsequently, when the temperature reaches 31.7 °C, the curve sharply rises to 41.30 °C and stabilizes. Throughout this change, two temperature intervals are considered suitable: −1.99 °C to −21.89 °C and 36.61 °C to 47.55 °C, with peaks at 3.37 °C and 41.30 °C, corresponding to suitability values of 0.93 and 1, respectively. The suitable range for bio14 (Figure 6d) is 23.64 to 109.83 mm, with the optimal value being 42.89 mm (details in Table 5).

### 3.5. The Centroid Variation in the Potential Distribution of P. fuscipes

The habitat of *P. fuscipes* in China under different climate scenarios is shown in Figure 7. In the SSP1-2.6 scenario, by the 2050s, the centroid of *P. fuscipes* moves southeastward from its current position by 618.41 km (111°71′ E, 28°31′ N), and by the 2090s, it moves northwestward by 42.87 km (111°47′ E, 28°66′ N). In the SSP2-4.5 scenario, from the current centroid position, it moves southeastward by 630.17 km to (112°07′ E, 29°10′ N) by the 2050s, and then continues moving southeastward by 361.92 km (112°09′ E, 25°49′ N). In the SSP5-8.5 scenario, from the current position to the 2050s and then to the 2090s, the centroid moves southeastward by 626.29 km (111°86′ E, 28°49′ N) and then continues moving southeastward by 11.92 km (111°97′ E, 28°44′ N). Under the three greenhouse gas concentration scenarios, the centroid displacement is relatively small from the 2050s to the 2090s in SSP1-2.6 and SSP5-8.5, while it is relatively larger in SSP2-4.5. Overall, under future climate scenarios, the centroid of *P. fuscipes* moves relatively uniformly southeastward compared to its current position (105°93′ E, 30°52′ N), except in the 2090s under the SSP1-2.6 scenario (see Table 6).

## 4. Discussion

This study utilized the MaxEnt model in conjunction with distribution data of *P. fuscipes* and key environmental variables to analyze its current suitable distribution locations in China [35]. Additionally, it also examined changes in suitability distribution under different future climate scenarios and explored the relationship between these changes and environmental variables. The model evaluation results indicated an AUC value of 0.968, suggesting a high level of accuracy in the model.

The prediction results of the MaxEnt model indicate that *P. fuscipes* has four distinct habitat suitability zones: high suitability, moderate suitability, low suitability, and unsuitable zones. The current distribution of suitable zones is relatively concentrated, with notably suitable zones in Xinjiang and Tibet. The high suitability zones are mainly concentrated south of the Sichuan Basin, north of the Yungui Plateau, in the southeast hills, the middle and lower reaches of the Yangtze River Plain, Hainan Province, and the northeastern direction of Taiwan Province, primarily focusing on the southern regions of China. Under future climate scenarios, the area of high suitability zones decreases significantly, and there is a trend of expansion and extension towards the coastal areas for suitable zones. *P. fuscipes* primarily inhabits humid environments such as riverbanks, marshes, and irrigated fields, which aligns with previous research findings [5]. Hunan is currently the province with the largest high suitability zone area in China, but under future climate scenarios, the high suitability zone area decreases while the areas of moderate and low suitability increase. Compared to North China, the northeast, and the northwest regions, East China and Central South China have the largest areas of high suitability. East China is located in the eastern part of China, characterized by hills, basins, and plains, with a subtropical monsoon climate a temperate monsoon climate, and abundant water resources. The climate conditions in Central and South China are similar to those in East China, mainly consisting of subtropical monsoons, temperate monsoons, and tropical monsoons. Annual rainfall ranges from 400 to 3000 mm, with annual accumulated temperatures exceeding 3000 °C. They are characterized by hot and rainy summers, humid and hot conditions, mild and wet winters, and distinct seasons. Analyzing Figure 4 and Table 4, over time, both the high and low suitability areas of *P. fuscipes* show a decreasing trend, with the most suitable habitats expanding towards the southeast. However, the high suitability areas are mainly concentrated in Shandong, Jiangxi, Anhui, Hubei, Henan, and Hebei provinces. With global climate warming, areas that were once suitable for *P. fuscipes* have become low suitability or unsuitable areas, while some unsuitable areas have become suitable habitats. In addition to climate factors affecting this insect, the distribution of various crops may also have an impact. Zhu et al., (1984) [8] pointed out that the distribution and occurrence of *P. fuscipes* in fields are closely related to the occurrence of pests in different crop types. *P. fuscipes* shifts with the occurrence time and quantity of various crop pests, which may be a reason for its future spread toward the southeast.

Temperature, precipitation, altitude, and other environmental variables have direct and indirect impacts on insect survival [36,37]. Temperature is an important environmental and survival factor for insects, including *P. fuscipes*. Insects are ectothermic animals, meaning their body temperature changes with environmental temperature fluctuations. These changes in body temperature can directly accelerate or inhibit metabolic processes, affecting their growth and reproduction [38]. This study used jackknife tests combined with Pearson correlation coefficients to select the important environmental variables that limit the distribution of *P. fuscipes,* which we identified as follows: bio11, bio9, bio14, bio7, bio6, bio2, bio3, bio13, and bio15. The results indicate that temperature and precipitation are significant factors affecting the distribution of *P. fuscipes*. The optimal temperature for the mean temperature of the coldest quarter is 6.06 °C; for the minimum temperature of the coldest month, it is −1.84 °C; for the mean temperature of the driest quarter, it ranges from 3.37 °C to 41.30 °C; and the optimal precipitation for the driest month is 42.89 mm. The suitability value curves for several major environmental variables show steep rises and falls, with narrow suitability value ranges. This may be one of the reasons why the suitability area of *P. fuscipes* is significantly reduced in future predictions [39,40]. Humidity, solar radiation, and air temperature affect the metabolic rates of animals and plants in tropical forests [41]. Previous studies have also shown that *P. fuscipes* prefers to overwinter in crop fields with dense vegetation cover that is relatively sheltered from the wind and cold. However, the humidity levels vary, with some areas being very humid and others even somewhat dry [8,42]. This also explains why *P. fuscipes* may appear in areas where it was previously unsuitable for growth in the future.

In addition to analyzing the current distribution, future distributions of *P. fuscipes* were also predicted. Under three typical greenhouse gas concentration scenarios, SSP1-2.6, SSP2-4.5, and SSP5-8.5, the area of highly suitable habitats for *P. fuscipes* in the 2050s (2041–2060) and 2090s (2081–2100) significantly decreased. Only a small amount of highly suitable area remained in a few areas such as Guangxi, Hunan, Fujian, Guangdong, and Guizhou, with the most pronounced decrease observed in the 2090s under the SSP1-2.6 scenario, with a reduction rate as high as 98.16%. Comparing greenhouse gas concentrations, the reduction in suitability was significantly greater under SSP1-2.6 than under SSP2-4.5 and SSP5-8.5, indicating that *P. fuscipes’* suitability is poorest under low greenhouse gas concentration environments. The areas of high suitability and low suitability are decreasing to varying degrees, while the area of moderate suitability is increasing to varying degrees. However, considering all scenarios for high, moderate, and low suitability areas, both the 2050s and 2090s show a significant reduction in suitable habitats, indicating a weakening trend in the future suitability of *P. fuscipes*. The expansion of suitable habitats is moving towards regions such as Xinjiang, Gansu, Liaoning, and Inner Mongolia, mainly located in the northwest, north, and northeast regions of China with a temperate continental climate characterized by cold winters and hot summers and significant annual and daily temperature variations. This may be related to climate warming caused by high CO^2^ emissions in the SSP5-8.5 scenario. Climate warming is causing temperatures to rise in traditionally cold regions like Inner Mongolia and Xinjiang, allowing *P. fuscipes* to expand into new territories. The current centroid of this insect is at 105°93′ E, 30°52′ N, but in the future, under different gas concentration scenarios, the centroid is projected to shift slightly southeastward from the 2050s to the 2090s, indicating no major directional shift in the suitability area of *P. fuscipes*.

Ecological indicators only describe the basic ecological requirements of a species and do not represent its actual ecological needs. When using species distribution models to predict potential distributions, various biotic and abiotic factors that influence species distribution are often overlooked [43,44,45]. The MaxEnt model used in this study is based on known distribution data for the target species and selected environmental data to study the species’ distribution probability. Although it has good predictive results, the selected environmental variables are limited, with only temperature and precipitation being considered, while other environmental factors are not included in the model. While the MaxEnt model has universal advantages in predicting potential species distributions, it also has limitations [27]. Response curves only show the impact of a single environmental factor, ignoring interactions between variables. Considering all environmental factors comprehensively in a specific model analysis is impractical, so treating this model as a fundamental niche model may be more effective [46]. As an insect, *P. fuscipes’* survival factors are influenced not only by its biological characteristics but also by crop distribution, predators, human activities, and other environmental factors [47,48]. Future studies on the suitable habitat of *P. fuscipes*, including both biotic and abiotic factors such as human activities and host types in the model, can improve the accuracy of model predictions.

## 5. Conclusions

Based on known distribution information and climate factors of *P. fuscipes*, MaxEnt modeling, and ArcGIS technology were successfully used to predict the current and future suitable habitat distribution of *P. fuscipes* in China. The results indicate that the main environmental variables influencing its distribution are bio11, bio9, bio14, bio7, bio6, bio2, bio3, bio13, and bio15. Under the current climate conditions, high suitability areas are mainly distributed south of the Sichuan Basin, north of the Yunnan-Guizhou Plateau, in the southeast hills, the middle and lower reaches of the Yangtze River Plain, Hainan Province, and the northeastern direction of Taiwan Province. In the 2050s and 2090s, under three typical greenhouse gas concentration scenarios (SSP1-2.6, SSP2-4.5, and SSP5-8.5), high suitability habitat areas for *P. fuscipes* significantly decreased. The decrease is most significant under the SSP1-2.6 low-concentration greenhouse gas scenario, and the predicted centroid position of potentially suitable habitat areas tends to move southeastward. This study provides a new perspective on the distribution and environmental impact factors of *P. fuscipes*, promotes its use as a biological control agent against other pests, and provides necessary references for its application in agricultural and forestry pest control.

## Figures and Tables

**Figure 1 insects-15-00437-f001:**
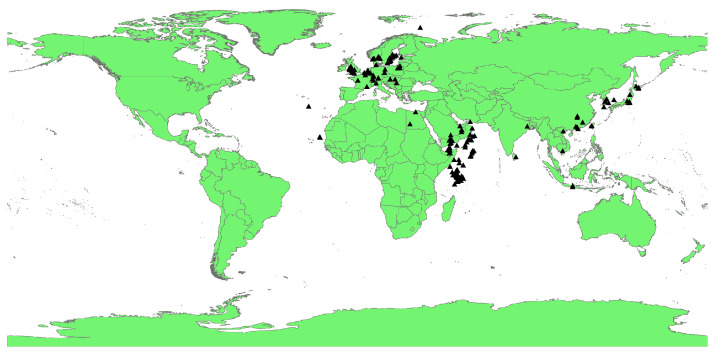
The global distribution records of *P. fuscipes*. (Black triangle: *P.fuscipes* distribution points).

**Figure 2 insects-15-00437-f002:**
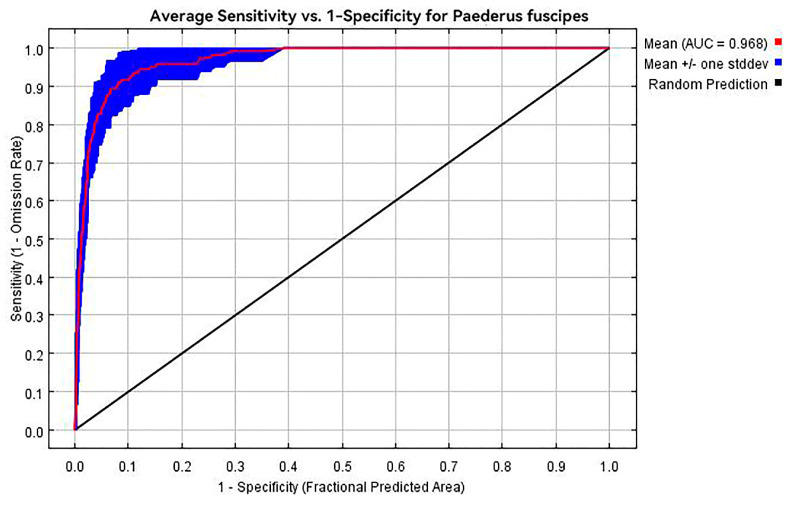
ROC curve of potential distribution modeling results for *P. fuscipes*.

**Figure 3 insects-15-00437-f003:**
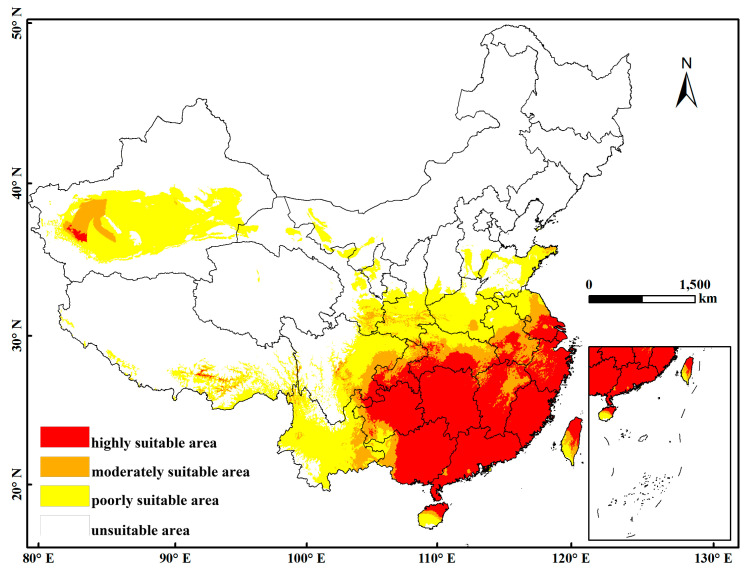
The suitable habitat distribution of *P. fuscipes* varies across different regions of China under the current climate conditions. (Red: high suitability zone; orange: moderate suitability zone; yellow: low suitability zone; white: unsuitable zone).

**Figure 4 insects-15-00437-f004:**
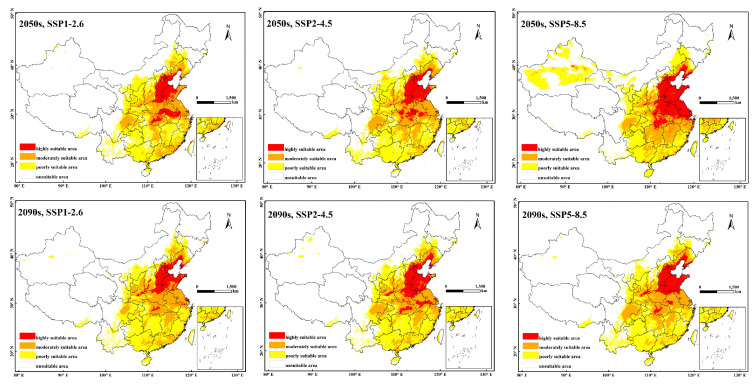
A predictive map of potentially suitable areas for *P. fuscipes* in China under different climate change scenarios.

**Figure 5 insects-15-00437-f005:**
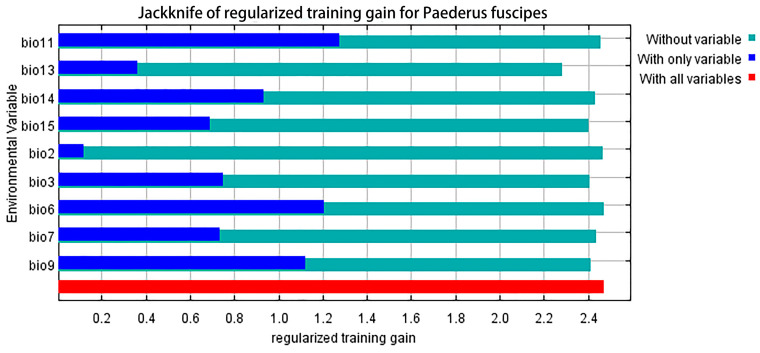
The jackknife test assesses the importance of environmental variables for *P. fuscipes*.

**Figure 6 insects-15-00437-f006:**
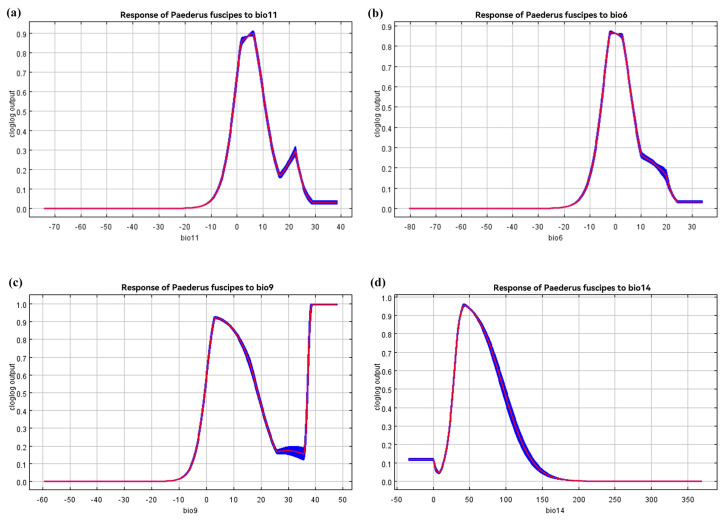
The probability of the presence of *P. fuscipes* and the response curve of key environmental variables.

**Figure 7 insects-15-00437-f007:**
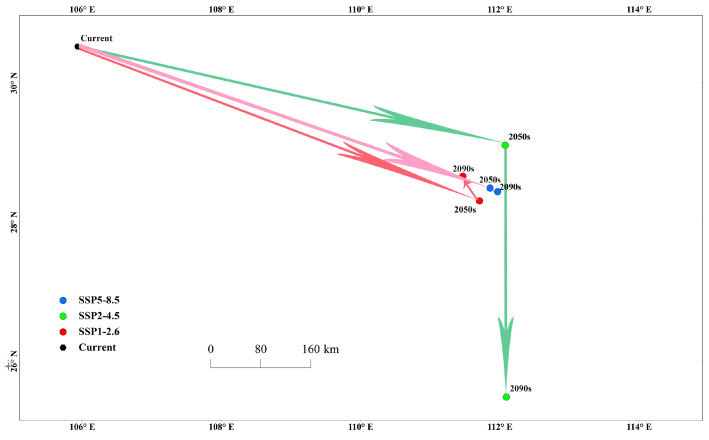
The change in the centroid of the potential distribution area of *P. fuscipes* in China.

**Table 1 insects-15-00437-t001:** The Pearson correlation coefficient between each environmental factor.

	bio2	bio3	bio6	bio7	bio9	bio11	bio13	bio14
**bio3**	0.776 **							
**bio6**	−0.208 *	−0.508 **						
**bio7**	0.519 **	−0.002	0.566 **					
**bio9**	0.061	−0.261 **	0.404 **	0.558 **				
**bio11**	−0.113	−0.393 **	0.627 **	0.520 **	0.647 **			
**bio13**	−0.015	0.030	0.182 *	0.257 **	0.301 **	0.143		
**bio14**	−0.329 **	−0.356 **	0.573 **	0.291 **	0.127	0.178 *	0.372 **	
**bio15**	0.705 **	0.680 **	−0.429 **	0.170	0.033	−0.164	0.153	−0.607 **

** Significant correlation at 0.01 level (bilateral). * Significant correlation at 0.05 level (bilateral).

**Table 2 insects-15-00437-t002:** Pearson’s correlation coefficients of crucial environmental variables.

Variable	Percent Contribution (%)	Permutation Importance (%)
Bio11	32	8.3
Bio9	22.7	8.6
Bio14	15.6	27.8
Bio7	9.8	3.5
Bio6	7.2	0.4
Bio2	4.4	10
Bio3	4.4	18.1
Bio13	2	17
Bio15	1.8	6.4

**Table 3 insects-15-00437-t003:** Predicted suitability for *P. fuscipes* in China under the current climatic conditions.

Province	High Suitable Area (10^4^ km^2^)	Total * (10^4^ km^2^)	Percentage of Highly Suitable Areas in the Province (%)	Percentage of Highly Suitable Areas in China (%)
Hunan	19.27	21.18	90.97	16.20
Guangxi	17.22	23.76	72.49	14.48
Guangdong	15.38	17.98	85.54	12.93
Jiangxi	12.03	16.69	72.11	10.12
Guizhou	11.98	17.62	68.01	10.07
Fujian	10.98	12.40	88.54	9.23
Zhejiang	8.32	10.60	78.50	7.00
Hubei	6.91	18.59	37.19	5.81
Chongqing	3.99	8.24	48.48	3.36
Anhui	3.19	14.01	22.79	2.68
Jiangsu	3.13	10.72	29.22	2.63
Sichuan	1.72	48.60	3.54	1.45
Taiwan	1.44	3.60	39.88	1.21
Hainan	1.06	3.54	29.82	0.89
Xinjiang	0.78	166.00	0.47	0.66
Shanghai	0.59	0.63	93.14	0.49
Yunan	0.56	39.41	1.42	0.47
Tibet	0.25	120.28	0.21	0.21
Hong Kong	0.09	0.11	81.06	0.08
Shaanxi	0.05	15.67	0.34	0.05
China	118.96	/	/	12.35

* Total area of the respective province.

**Table 4 insects-15-00437-t004:** Predicted suitable areas for *P. fuscipes* under the current and future climatic conditions.

Decade Scenarios	Predicted Area (10^4^ km^2^)			Comparison with Current Distribution (%)		
	High Suitable	Medium Suitable	Low Suitable	High Suitable	Medium Suitable	Low Suitable
current	118.96	50.73	168.05	—	—	—
2050s, SSP1-2.6	8.09	86.87	107.19	−93.20%	71.22%	−36.22%
2050s, SSP2-4.5	17.79	79.09	104.27	−85.04%	55.89%	−37.95%
2050s, SSP5-8.5	13.93	72.27	103.18	−88.29%	42.45%	−38.60%
2090s, SSP1-2.6	2.19	71.41	149.93	−98.16%	40.76%	−10.78%
2090s, SSP2-4.5	6.14	49.38	135.11	−94.84%	−2.67%	−19.60%
2090s, SSP5-8.5	27.64	63.10	108.52	−76.76%	24.38%	−35.42%

**Table 5 insects-15-00437-t005:** The optimal range of environmental variables corresponds to the potential distribution of *P. fuscipes*.

Environmental Variables	Suitable Range	Optimum Value
Bio11/°C	−2.83~−13.32	6.06
Bio6/°C	−7.30~−9.33	−1.84
Bio9/°C	−1.99~21.89; 36.61~47.55	3.37; 41.30
Bio14/mm	23.64~109.83	42.89

**Table 6 insects-15-00437-t006:** Under climate change scenarios, the centroid displacement trajectory of *P. fuscipes*-suitable habitats.

Scene	Period	Angle/°	Direction	Displacement/km
**SSP1-2.6**	Contemporary to 2050s	291.02	Southeast	618.41
	2050s to 2090s	145.76	Northwest	42.87
	Contemporary to 2090s	288.62	Southeast	583.69
**SSP2-4.5**	Contemporary to 2050s	283.01	Southeast	630.17
	2050s to 2090s	359.72	Southeast	361.92
	Contemporary to 2090s	309.29	Southeast	795.59
**SSP5-8.5**	Contemporary to 2050s	288.97	Southeast	626.29
	2050s to 2090s	295.25	Southeast	11.92
	Contemporary to 2090s	289.08	Southeast	638.15

## Data Availability

The data supporting the results are available in a public repository at: GBIF.org (19 February 2024) GBIF Occurrence Download https://doi.org/10.15468/dl.jatd47, accessed on 19 February 2024; *P. fuscipes* occurrence data: https://doi.org/10.6084/m9.figshare.25242757.v1, accessed on 19 February 2024.

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
