# Peer review of "Predicting the Impact of Climate Change on the Future Distribution of Paederus fuscipes Curtis, 1826, in China Based on the MaxEnt Model"

_insects, 2024, doi:10.3390/insects15060437_

Round 1

Reviewer 1 Report (Previous Reviewer 1)

Comments and Suggestions for Authors

This version is significantly better than the previous one. There are, however, still some issues which prevent me from recommending it for publication in the current form. I attach the file with my comments here.

Comments on the Quality of English Language

It reads well and only in a few cases, something should be improved (indicated in the file).

Author Response

Dear Reviewer:

Thank you for your comments concerning our manuscript entitled " Predicting the impact of climate change on the future distribu-tion of Paederus fuscipes Curtis, 1826, in China based on the MaxEnt model" (insects-3028544). Those comments are all valuable and very helpful for revising and improving our paper, as well as the important guiding significance to our research. We have studied the comments carefully and have made corrections which we hope to meet with approval. The main corrections in the paper and the response to the your comments are as flowing:

For Reviewer 1

Minor comments,

1)" The genus and species names should be in italics. Also, give the name of the author and a year after the genus name when mentioned first time in the text.(line 45)

Author's response: Thanks for your meaningful comment. The genus name is modified to italic in this paper. (Line 45-46, in the revised manuscript)

2)" 'vector', I think it would be beneficial for the readers to add the information on the name of the substance and the bacterium, which is producing it.(line 47)

Author's response: Thanks for your meaningful comment. The word "vector" is described in detail in the original text. (Line 48-50, in the revised manuscript)

3). " Why would you consider Paederus as pest? It was just said that it is an enemy of pests.(line 48)

Author's response: Thanks for your meaningful comment. I'm very sorry; this was my oversight. "pest" has been changed to "insect." (Line 51 in the revised manuscript)

4). " 'Globally, there are approximately over 65,000 species of rove beetles,' This information is not needed.(line 50)

Author's response: Thanks for your meaningful comment. This sentence has been deleted from the original text. (Line 52-53, in the revised manuscript)

5). " This rather applies to Paederus than all Staphylinidae.(line 51)

Author's response: Thanks for your meaningful comment. The text has been amended accordingly. (Line 52-54, in the revised manuscript)

6). " 'P. fuscipes' In italics.(line 52)

Author's response: Thanks for your meaningful comment. "P. fuscipes" has been italicized. (Line 55, in the revised manuscript)

7). " 'isn't' changed to 'is not'(line 54)

Author's response: Thanks for your meaningful comment. 'isn't' has been changed to 'is not'. (Line 57, in the revised manuscript)

8). " 'diverse diet' As far as we know, Paederus beetles are predators, no different diet.(line 56)

Author's response: Thanks for your meaningful comment. Deleted "diverse diet", and the corresponding parts have been revised. (Line 59, in the revised manuscript)

9). " delete 'decaying organic matter'(line 57)

Author's response: Thanks for your meaningful comment. "decaying organic matter" has been deleted. (Line 60, in the revised manuscript)

10). " 'P. fuscipes' In italics.(line 58)

Author's response: Thanks for your meaningful comment. "P. fuscipes" has been italicized. (Line 61, in the revised manuscript)

11). " Why is P. tamulus suddenly brought here?(line 58-59)

Author's response: Thanks for your meaningful comment. The introduction of "P. tamulus" was meant to illustrate that both types of rove beetles can prey on these pests. "P. tamulus" has now been removed from the text. (Line 61-62, in the revised manuscript)

12). " 'P. fuscipes' In italics.(line 75, 77, 78, 81)

Author's response: Thanks for your meaningful comment. "P. fuscipes" has been italicized. (Line 78, 80, 81, 84, in the revised manuscript)

13). " The address is missing here.(line 91)

Author's response: Thanks for your meaningful comment. The address has been completed. (Line 95, in the revised manuscript)

14). " What about the record from Australia? Check Atlas of Living Australia(line 97)

Author's response: Thanks for your meaningful comment. We did not find any distribution points for Australia in GBIF or the literature, so no distribution points for Australia are shown in the map.

15). " 'Using ArcGIS to extract the distribution areas of P. fuscipes in China and analyze the climatic suitability of this insect. '. Verb is missing in this sentence.(line 125)

Author's response: Thanks for your meaningful comment. The sentence has been modified. (Line 129-130, in the revised manuscript)

16). " ' This suggests that P. fuscipes has strict requirements for temperature and precipitation, with narrow suitability ranges and weaker suitability abilities in the environment. '. Can we really say that about a species that occurs at four continents? For me, it looks like it is not really narrow range of suitability.(line 324-326)

Author's response: Thanks for your meaningful comment. The sentence has been modified. (Line 328-330, in the revised manuscript)

17). "Why is Stenus appearing suddenly?(line 332)

Author's response: Thanks for your meaningful comment. The sentence has been modified. (Line 336-337, in the revised manuscript)

18). "What is "host" distribution in this context?(line 373)

Author's response: Thanks for your meaningful comment. The word "host" has been deleted, and the sentence has also been modified. (Line 375-378, in the revised manuscript)

We tried our best to improve the manuscript and made some changes in the manuscript. These changes will not influence the content and framework of the paper.

We appreciate for your warm work earnestly and hope that the correction will meet with approval.

Once again, thank you very much for your comments and suggestions.

Regards,

Xinju Wei

College of Life Science, China West Normal University,

1 Shida Road, Nanchong, 637002, China

[Email] weixinjuxx@foxmail.com

Reviewer 2 Report (Previous Reviewer 2)

Comments and Suggestions for Authors

The authors do not pay attention to the mandatory writing of species names in italics.

To the extent possible, I pointed out these errors.

There are still minor technical errors in the text.

As I wrote for the first time, the distribution of the species is not fully described in GBIF, https://www.gbif.org.

However, the analysis of the presented data is very convincing and allows us to accept conclusions in this case.

Line 23 Paederus must be Paederus Fabricius, 1775.

line 45  Paederus - must be italics and Paederus Fabricius, 1775.

lines 47, 52, 75, 77, 78, 79, 81, 87, 92, 116, 118, 125, 129, 133, 134, 152, 161, 163, 167,  174,  186,  192, 195, 201, 203, 212-213, 215, 217,  220, 221, 225, 232, 233, 248, 250, 253,  255, 256, 268, 269, 272-273, 279, 287, 300, 307, 312, 317,  319, 324, 329, 332-334, 336, 338, 344, 349, 359, 372, 375,  378, 380, 415     P. fuscipes - must be italics

line 58-59  P. fuscipes - Paederus tamulus must be italics

line 202, 208, km2

Table 4 in title of column: km2

Table 4 in title of columns: into two lines!

Figure 5, 6: Paederus fuscipes must be italics

line 322: must be 48.89 mm

line 332: Stenus - must be italics

line 354: must be CO2

line 430, 435, 442, 480, 488, : - must be italics

Author Response

Dear Reviewer:

Thank you for your comments concerning our manuscript entitled " Predicting the impact of climate change on the future distribu-tion of Paederus fuscipes Curtis, 1826, in China based on the MaxEnt model"(insects-3028544). Those comments are all valuable and very helpful for revising and improving our paper, as well as the important guiding significance to our research. We have studied the comments carefully and have made corrections which we hope to meet with approval. The main corrections in the paper and the response to your comments are as flowing:

For Reviewer 2

Comments and Suggestions for Authors,

  • " Line 23 Paederus must be Paederus Fabricius, 1775.”

Author's response: Thanks for your valuable comment. The genus name is modified to italic in this paper. (Line 23-24, in the revised manuscript)

  • " line 45 Paederus - must be italics and Paederus Fabricius, 1775.”

Author's response: Thanks for your valuable comment. The genus name is modified to italic in this paper. (Line 45-46, in the revised manuscript)

  • " lines 47, 52, 75, 77, 78, 79, 81, 87, 92, 116, 118, 125, 129, 133, 134, 152, 161, 163, 167, 174, 186,  192, 195, 201, 203, 212-213, 215, 217,  220, 221, 225, 232, 233, 248, 250, 253,  255, 256, 268, 269, 272-273, 279, 287, 300, 307, 312, 317,  319, 324, 329, 332-334, 336, 338, 344, 349, 359, 372, 375,  378, 380, 415  fuscipes - must be italics

Author's response: Thanks for your valuable advice. "P. fuscipes" has been italicized.

  • "line 58-59 fuscipes - Paederus tamulus must be italics”

Author's response: Thanks for your valuable advice. "P. fuscipes" has been italicized. (Line 61, in the revised manuscript)

  • "line 202, 208, km2

Author's response: Thanks for your valuable advice. "km2" has been changed to "km2". (Line 206, 212, in the revised manuscript)

  • Table 4 in title of column: km2

Author's response: Thanks for your valuable advice. "km2" has been changed to "km2".

  • Table 4 in title of columns: into two lines!

Author's response: Thanks for your valuable advice. Table 4 has been amended.

  • Figure 5, 6: Paederus fuscipes must be italics

Author's response: Thanks for your meaningful comment. "P. fuscipes" has been italicized. (Line 252, 254, in the revised manuscript)

  • line 322: must be 48.89 mm

Author's response: Thanks for your meaningful comment. “48.89mm” has been modified to “48,89 mm”. (Line 326, in the revised manuscript)

  • line 332: Stenus - must be italics

Author's response: Thanks for your meaningful comment. The sentence has been modified. (Line 336-337, in the revised manuscript)

  • line 354: must be CO2

Author's response: Thanks for your meaningful comment. “CO2” has been modified to “CO2”. (Line 357, in the revised manuscript)

  • line 430, 435, 442, 480, 488, : - must be italics

Author's response: Thanks for your meaningful comment. The corresponding location has been revised.

We tried our best to improve the manuscript and made some changes in the manuscript. These changes will not influence the content and framework of the paper.

We appreciate for your warm work earnestly and hope that the correction will meet with approval.

Once again, thank you very much for your comments and suggestions.

Regards,

Xinju Wei

College of Life Science, China West Normal University,

1 Shida Road, Nanchong, 637002, China

[Email] weixinjuxx@foxmail.com

This manuscript is a resubmission of an earlier submission. The following is a list of the peer review reports and author responses from that submission.

Round 1

Reviewer 1 Report

Comments and Suggestions for Authors

I generally like the idea of such study, but the manuscript needs significant improvement. It seems that the authors do not really have much knowledge on the studied animal and much of the general information is simply incorrect. Please, see my comments and corrections directly in the file. In my opinion, the manuscript is not ready for publication and I recommend a rejection. But at the same time, I would highly encourage the authors to improve their knowledge and the manuscript and resubmit it in the new form.

Comments on the Quality of English Language

English is generally ok, but in some cases the sentences are confusing. I marked some of those in the file. Also, some of the sentences are too long and/or too repetitive.

Reviewer 2 Report

Comments and Suggestions for Authors

The article under review makes an ambivalent impression.

I have no comments about the mathematical apparatus used. Predicting climate change is very interesting.

However, what is associated with the Paederus fuscipes is extremely unfortunate and largely incorrect.

In title (line 3) must be fuscipes fuscipes Curtis, 1826.

And it is a first problem.

Let's start with the fact that two subspecies of Paederus fuscipes are known in China: fuscipes fuscipes Curtis, 1826 and fuscipes sinensis Scheerpeltz, 1957. Moreover, in the province of Sichuan (Szechwan) they are found together.

Paederus fuscipes belongs to the order Coleoptera, the family Staphylinidae, and the genus Paederus” (line 11).

Must be “Paederus fuscipes belongs to the order Coleoptera, the family Staphylinidae.

„hidden-winged beetle” (line 14). Must be rove beetle.

Paederus genus (line 22). Must be: the genus of Paederus Fabricius, 1775

Paederus fuscipes represents a species within the Paederus genus found in Fujian, Guangdong, Hainan, and Guizhou, among other locations [2]. (line 54-55). Must be: Paederus fuscipes found in Fujian, Guangdong, Hainan, and Guizhou, among other locations [2]

The distribution of P. fuscipes in China is written very roughly: East and South Central provinces (line 34).

The description of morphology (lines 45-54) is superficial and generally not necessary in this kind of article.

body fluid (lines 58-59). Must be hemolymph.

Not Paederus toxin, but Pererine toxin (line 59).

Paederus fiscipes is not a parasite (line 63)

Australia, France, and Korea (line 101). What about: Northern Iran, Nigeria, France, Okinawa, Australia, Malaysia, Indonesia, Thailand, Singapore, Taiwan, Vietnam, India (Perumbavoor, Kerala), Sierra Leone, Sri Lanka and Ethiopia  (https://en.wikipedia.org/wiki/Paederus_dermatitis)

The GBIF website (lines 102-104), http://www.gbif.org/, does not list known finds of the beetle in the provinces Fujian (Fukien), Sichuan (Szechwan), Sichuan Yunnan, Chongqing, Guizhou (Kweichow), and  Guanxi (Kwangsi). This location are from the “Catalogue of Palaearctic Coleoptera (Catalogue of Palaearctic Coleoptera. Hydrophiloidea-Staphylinoidea. Revised and updated edition. Volume 1 (2015). Edited by I. Löbl, D. Löbl. BRILL, Leiden, Boston. 1730 p. ISBN 978-90-04-28992-5

The “Catalog” information does not include geographic coordinates. However, these are geographical points.

About the geographical range of the Paederus fiscipes. The Paederus fiscipes is distributed over all continents (except Antarctica) and large islands. Calling it invasive and spreading is incorrect. It has already settled in suitable habitats. This beetle is a hygrophilous, an inhabitant of open spaces: fields, meadows, marshes and the coasts of fresh water bodies. So its settlement in China to the west and northwest under the influence of climate change is impossible. This is actually shown on the map (fig. 3)

The westernmost point (105°93'E, 30°52'N) is in Sichuan, but there is a find in Yunnan not shown in GBIF. This find is much further south.

Within the predicted boundaries (112°E) the this species is found now.